# Molecular Basis of Endometriosis and Endometrial Cancer: Current Knowledge and Future Perspectives

**DOI:** 10.3390/ijms22179274

**Published:** 2021-08-27

**Authors:** Milan Terzic, Gulzhanat Aimagambetova, Jeannette Kunz, Gauri Bapayeva, Botagoz Aitbayeva, Sanja Terzic, Antonio Simone Laganà

**Affiliations:** 1Department of Medicine, School of Medicine, Nazarbayev University, Kabanbay Batyr Avenue 53, Nur-Sultan 010000, Kazakhstan or terzicmm@pitt.edu (M.T.); sanja.terzic@nu.edu.kz (S.T.); 2National Research Center for Maternal and Child Health, Clinical Academic Department of Women’s Health, University Medical Center, Turan Avenue 32, Nur-Sultan 010000, Kazakhstan; gauri.bapaeva@gmail.com (G.B.); aitbayeva_botagoz@inbox.ru (B.A.); 3Department of Obstetrics, Gynecology and Reproductive Sciences, University of Pittsburgh School of Medicine, 300 Halket Street, Pittsburgh, PA 15213, USA; 4Department of Biomedical Sciences, School of Medicine, Nazarbayev University, Kabanbay Batyr Avenue 53, Nur-Sultan 010000, Kazakhstan; jeannette.kunz@nu.edu.kz; 5Department of Obstetrics and Gynecology, “Filippo Del Ponte” Hospital, University of Insubria, 21100 Varese, Italy; antoniosimone.lagana@uninsubria.it

**Keywords:** endometriosis, endometrial cancer, ovarian cancer, molecular basis of endometriosis

## Abstract

The human endometrium is a unique tissue undergoing important changes through the menstrual cycle. Under the exposure of different risk factors in a woman’s lifetime, normal endometrial tissue can give rise to multiple pathologic conditions, including endometriosis and endometrial cancer. Etiology and pathophysiologic changes behind such conditions remain largely unclear. This review summarizes the current knowledge of the pathophysiology of endometriosis and its potential role in the development of endometrial cancer from a molecular perspective. A better understanding of the molecular basis of endometriosis and its role in the development of endometrial pathology will improve the approach to clinical management.

## 1. Morphological Features of the Human Endometrium

The uterine endometrium is an inner mucosal layer of the uterine cavity with the unique ability to regenerate or shed depending on the phases of the menstrual cycle and hormonal levels [1,2]. The human endometrium consists of two layers: functional (stratum functionalis) and basal (stratum basalis). The endometrium undergoes structural modification and changes in specialized cells in response to fluctuations of estrogen and progesterone during the menstrual cycle [3]. The basal layer of the endometrium is responsible for the regeneration of functional layer during the proliferative phase [4,5,6]. A hypothesis on the regeneration process of the endometrium suggests that the functional layer quality depends on endometrial progenitors/stem cells located in the basal layer [7,8,9]. However, understanding of the regenerative mechanism of the endometrium during the menstrual cycle and the location of endometrial progenitor/stem cells have not been fully elucidated [10,11,12]. The traditional morphological theory of the endometrium describes it as two-dimensional (2D) histological structure [13,14,15]. However, due to the complexity of the morphology of the endometrial glands, the technical characteristics of 2D histopathological imaging have been found to be insufficient [4].

It was hypothesized that clonal genomic alterations in histologically normal endometrial glands may change the stereoscopic structure of the endometrial glands. Three-dimensional (3D) pathological morphology of tissue affected by adenomyosis and 3D morphology of the normal endometrial glands was compared using 3D full-thickness images of the human uterine endometrium with microscopy [4]. 3D imaging revealed a more complex network of endometrial glands in human endometrium than was observed with traditional 2-dimensional (2D) imaging [4]. Using 3D imaging, Yamaguchi and co-authors (2021) found specific morphological features of human endometrial glands, including occluded glands, the plexus of the basal glands, and the gland-sharing plexus with other glands, which were not observed in the past using 2D histological methods [4]. The 3D analysis of the endometrial layers clarified that the plexus structure of the glands expanded horizontally along the muscular layer. Furthermore, these morphological features were detected regardless of age or phase of the menstrual cycle, suggesting that they are basic components of the normal human endometrium [4]. These novel findings suggest that 2D histology, which has been in use for more than 100 years, does not adequately depict the morphology of the endometrium. A clearer picture of the structure of the human could develop our understanding of various endometrial conditions and the etiology of endometriosis and endometrial cancer (EC). These diseases significantly affect reproductive age women and impact their quality of life [16,17,18]. Understanding the pathogenesis, immunohistochemical and molecular mechanisms of these conditions could improve the management of patients with endometriosis and EC [19,20,21,22].

## 2. Endometriosis

### 2.1. Definition, Epidemiology and Classification

Endometriosis is an estrogen-dependent inflammatory disorder of the endometrium that is characterized by the presence of functionally active endometrial tissue, stroma and glands outside the uterine cavity [21,23,24,25,26]. This condition estimated to affect up to 11% of women in reproductive age (or ∼200 million women) worldwide and up to 50% of women with pelvic pain or infertility [21,24,25,27,28]. The etiology of endometriosis is largely unknown. Previous research has shown that endometriosis is prevalent after menarche and dramatically drops after menopause, which has led researchers to believe that the disorder is estrogen- and progesterone-dependent [26,27,29].

There are different classifications of endometriosis based on staging and types (Table 1) [30]. According to the revised American Society for Reproductive Medicine (ASRM) scoring system [31,32], endometriosis is classified into four stages based on the localization and extension of the implants. The disease is classified as peritoneal, ovarian, or deep infiltrating endometriosis, which can be roughly described as the presence of endometrial tissue expanding to a depth of more than 5 mm below the peritoneum [22,32,33]. The classification includes four stages based on the severity, quantity, location, depth, and size of growths, those stages being stage I (minimal disease), stage II (mild disease), stage III (moderate disease), and stage IV (severe disease) [26,33,34]. This classification, however, has not been shown to be a reliable predictor of clinical outcomes.

As the supplement to the ASRM classification, and in order to provide a morphologically descriptive classification of deep infiltrating endometriosis, the ENZIAN classification was developed (Table 1) [30]. It takes into account retroperitoneal structures.

The Endometriosis Fertility Index (EFI) is another attempt to improve the endometriosis classification (Table 1). The EFI aims to predict pregnancy rates in patients with surgically documented endometriosis who attempt non-IVF conception. The EFI classification is a scoring system that includes assessment of factors related to a patient’s history at the time of surgery, of adnexal function at conclusion of surgery, and of the extension of endometriosis [30].

Following another classification, endometriosis is subdivided into three types: superficial peritoneal disease, ovarian endometrioma, and deep endometriotic lesions [35,36]. Adenomyosis, as “internal” uterine endometriosis, is characterized by the presence of endometrial glands and stromas within the myometrium that causes myometrial inflammation and hypertrophy [35,37,38]. Adenomyosis can be classified in several different subtypes: (a) intrinsic adenomyosis, (b) extrinsic adenomyosis, (c) adenomyosis externa, and (d) focal adenomyosis located in the outer myometrium [35,37,38]. Although there are many studies supporting this new classification, international consensus has not yet been achieved [35].

A major disadvantage of all existing classifications is that no one of them links the severity of the pain with the findings (imaging, laparoscopic) [39]. Some patients who are classified as “severe” by ASRM experience little pain but have associated infertility. Others, with only superficial red and blue lesions and minor adhesions, may experience severe pain and consequently a low quality of life [39,40,41].

Abrao and Miller recently proposed a new classification system [42]. They propose that a classification should (1) clearly describe the sites and extent of disease; (2) provide a close correlation with the symptoms of endometriosis; (3) reflect the surgical difficulty encountered relative to the disease location; (4) be user-friendly with tools that are conducive to support a surgeon’s busy practice by enabling completion of documentation immediately upon procedure conclusion; (5) be validated for both pain and infertility; (6) create a comprehensive universal language that is meaningful for clinical practitioners and researchers [39,42].

### 2.2. Risk Factors of Endometriosis

A number of modifiable and non-modifiable risk factors have been reported to be both positively and negatively associated with the development of endometriosis (Figure 1) [27,28,43,44]. Non-modifiable risk factors known to be associated with endometriosis are the following: genetic, endocrine, immunological, and ethnicity [21,45]. There are also modifiable factors, the effect of which could be decreased substantially by lifestyle changes. Those factors are microbiotic, environmental factors (exposure to endocrine-disrupting chemicals), alcohol/caffeine intake, smoking, and physical activity [27]. Those factors may influence estrogen levels and contribute to the development of endometriosis [27].

The risk of endometriosis has been strongly linked to ethnicity. Many researchers have reported a nine-fold increase in risk of endometriosis development among women of Asian ethnicity if compared with the European-American Caucasian female population [27,43,46]. Among other factors, prolonged estrogen exposure (e.g., early age at menarche, shorter menstrual cycles, nulliparity) [47], low body mass index, and uterine outlet obstruction [48] have been suggested as predisposing to endometriosis.

It is well known that endometriosis has a strong genetic predisposition [25,43]. The evidence for an association between genetic polymorphisms and risk of endometriosis is robust [43]. Together with the strong link to hereditary factors, development of endometriosis is also affected by environmental exposures [26]. Environmental factors such as elevated levels of phthalate esters, persistent organochlorine pollutants, perfluorochemicals, and exposure to cigarette smoke can increase risk of developing endometriosis by inducing oxidative stress, altering hormonal homeostasis, or by changing immune responses [43]. Maternal exposure to diethylstilbestrol (DES) has been associated with a greater risk of endometriosis in female offspring [28].

Modifiable risk factors such as caffeine intake have been hypothesized to be influential in the pathology of gynecological disease due to its ability to influence estradiol levels [27,49]. Much like caffeine, alcohol intake and tobacco smoking are hypothesized to alter reproductive hormones due to the activation of aromatases leading to increased conversion of testosterone to estrogens [27]. Moreover, tobacco smoking may also increase the inflammatory response. Physical activity has been shown to reduce the risk of developing many gynecological diseases [27,50]. Other risk factors, such as the presence of lower genital tract infections, have also been proposed as risk factors. [43,51].

Some genetic factors have been found to serve as risk factors for endometriosis. Genome-wide association studies have, to date, identified 19 independent single-nucleotide polymorphisms (SNPs) as being significantly associated with endometriosis [52]. Moreover, the authors found a significant genetic overlap between endometriosis and EC in a genetic correlation analysis, which found 13 SNPs that appeared to be involved in development of both conditions [52].

### 2.3. Pathophysiology of Endometriosis

To date, the etiology and pathogenesis of endometriosis remains controversial. Multiple theories have been proposed to explain the pathogenesis of endometriosis [24,25,38]. Among the most recognized and reasonable are retrograde menstrual blood flow, coelomic metaplasia, and Müllerian remnants theories [21,24,39,53,54]. Amongst the various hypotheses, the one that has the greatest consensus is Sampsons’ retrograde menstruation. Retrograde menstruation is the process in which endometrial cells and fragments of the tissue shed during menstrual bleeding and are transported into the peritoneal cavity due to the retroperistaltic movements of the fallopian tubes [21,24,53]. Implantation of these particles and subsequent proliferation during the menstrual cycle leads to the damage of pelvic organs at positions of implantation [21]. However, the hypothesis about the retrograde menstruation as a potential cause of endometriosis does not explain localization of endometrial tissue that can be found in rare cases of extragonadal endometriosis and endometriosis in male patients [26]. Another theory suggests that endometriosis develops due to endometrial cells transferred through the lymphatic system to other parts of the body, where they further grow and proliferate [21,24,26]. Additionally, it has been proposed that circulating blood cells originating from bone marrow differentiate into endometriotic tissue at various body sites [24,55]. Distant organ endometriosis, such as lung and brain endometriosis, is very rarely described and might be explained by vascular spread [21,24].

Meyer’s hypothesis about coelomic metaplasia suggests development of endometriosis from the visceral epithelium, which can be converted to endometrial tissue by metaplastic processes [21,39].

More recent studies suggest that endometriosis is a pelvic inflammatory condition, so called “peritonitis without germs” [39,53]. This is based on the fact that the peritoneal fluid has an increased concentration of activated macrophages and an inflammatory profile in the cytokine/chemokine axis [39,56]. Cousins and Gargett in 2018 proposed that the human endometrium regenerates cyclically every month mediated by endometrial stem/progenitor cells such as CD140b+, CD146+, or SUSD2+ endometrial mesenchymal stem cells (eMSCs) [57]. N-cadherin + endometrial epithelial progenitor cells and side population cells may also contribute to the pathogenesis of the disease. They hypothesized that the eMSCs may have a role in the generation of progesterone-resistant phenotype endometrial stromal fibroblasts [39,57]. According to the other recent theories, deregulation of genes and the Wingless-related integration site (Wnt)/β-catenin signaling pathway would produce an aberration and the axial extension of the identity of the anterior-posterior patterning, whilst a deregulation of Hox genes and cofactor pre-B-cell leukemia homeobox 1 (Pbx1) produces an aberration in the segmentation of the mesoderm [21]. This may cause aberrant placement of stem cells with endometrial phenotype and maintain them in a quiescent niche. Transcriptional activity induces the expression of vascular endothelial growth factor (VEGF) that stimulates the vascular endothelial cell. On the other hand, Müllerian inhibiting factor (MIF) induces endometrial cell mitosis, whose survival is supported by the activation of antiapoptotic gene B-cell lymphoma 2 (Bcl-2), by the degradation of the extracellular matrix by matrix metalloproteinases (MMPs) via intercellular adhesion molecule 1 (ICAM-1) and vascular cell adhesion protein 1 (VCAM-1), creating the conditions for differentiation, adhesion, proliferation, and survival of ectopic endometrial cells [21]. This will lead to decreased apoptosis of ectopic endometrial-like cells [58,59,60], which escape from immune surveillance, and subsequently implant and proliferate. According to a recent review by Patel and colleagues, there is growing evidence that hormonal and immune factors create a pro-inflammatory microenvironment that support the persistence of endometriosis [61]. It is clear there is still much to learn about the nature and pathophysiology of endometriosis, and development of these theories could contribute to a greater understanding of the disease.

### 2.4. Clinical Presentation and Diagnostic Tools

Endometriosis is difficult to diagnose for many reasons: lack of clear understanding of etiologic factors, diversity of hypotheses for pathogenesis, different clinical presentation of the disease, and existence of asymptomatic cases [62]. Careful patient interview including family history, detailed examination, and additional imaging work-up are required for diagnosis [63,64].

Most women diagnosed with endometriosis present with multiple diverse symptoms [25]. Commonly reported complaints include chronic pelvic pain, dysmenorrhea, dyspareunia, dyschezia, and infertility/subfertility [25,33,39]. 

Chronic pelvic pain accounts for 10% of outpatient gynecologic visits, while local pain or tenderness on pelvic examination is associated with pelvic disease in 97% of patients and with endometriosis in 66% of patients [65]. Dysmenorrhea and general pelvic pain are common symptoms of endometriosis, regardless of age at diagnosis [66]. Pelvic pain due to endometriosis is usually chronic (lasting ≥6 months) and is associated with dysmenorrhea (in 50 to 90% of cases), dyspareunia, deep pelvic pain, and lower abdominal pain with or without back and loin pain [65]. Most women experience pain of different severity: from mild or moderate pain (pain usually requiring medication) to severe pain (pain requiring medications and bed rest) during menses over the lifetime [66]. Pain in endometriosis has a complex mechanism. Increased systemic and local proinflammatory cytokines and growth factors due to the chronic inflammation in endometriosis contribute to the mechanism of chronic pain development through persistent noxious stimulation, chronic inflammation, and nerve injury, which will alter pain processing and result in central sensitization [25,62]. Surgical treatment in many cases increases central sensitization, and patients often report worsening of symptoms after surgery [25,67]. The severity of pain is often associated with the depth of endometriotic infiltration rather than the size of the lesion or cyst [25,62,68]. Dyspareunia is another common symptom that is closely related to pain and nerve sensitization [25]. 

Some patients may experience gastrointestinal (nausea and vomiting, more frequent bowel movements accompanying pelvic pain) and urinary (frequent urination when experiencing menstrual pain) symptoms [65,66].

Infertility and subfertility are other important issues related to endometriosis. In cases of severe and deep infiltrating endometriosis [22,33,69], the mechanism of infertility is the alteration of normal anatomy of the reproductive organs [25]. However, in cases of a small ectopic endometrial implants/lesions, the mechanism of infertility is not clear yet. The authors suggested an endometrial defect as the explanation of implantation impairment in endometriosis. This hypothesis is supported by numerous studies showing decreased expression of several biomarkers of implantation [25,69].

Following the key steps during the initial clinical examination in the diagnosis of women with endometriosis, imaging investigations should be done in order to confirm the condition. Some biological tests invented currently have little or no merit in the diagnosis of endometriosis, and no biomarker tests have been identified to be conclusive [26,62,70,71]. In contrast, imaging techniques led to substantial improvements in the diagnosis of endometriosis [25,62,72]. The most helpful tools are transvaginal ultrasound (TVUS) [73,74] and MRI [62,72]. In addition, sigmoid, ileocecal, and urological lesions can be detected with supplementary radiological techniques such as transrectal sonography (TRS), rectal endoscopic sonography (RES) [75,76], multidetector CT scan with retrograde colonic opacification and late urography, and/or uro-MRI [62,77]. However, a recent Cochrane meta-analysis reported inconclusive data from TRS and RES studies [77]. If using these methods, it is important to remember that TRS (5 MHz frequency) enables a limited analysis of the rectosigmoid colon, whereas RES (7.5–12 MHz) provides an overview of the whole sigmoid and rectosigmoid colon with higher spatial resolution [72].

## 3. Endometrial Cancer

### 3.1. Definition, Epidemiology and Classification

Endometrial cancer is a malignant disease of the inner layer of the uterus (endometrium) [3,78]. It is one of the most common gynecological malignant tumors in developed countries [3,78,79,80]. In 2012, 527,600 women worldwide were diagnosed with EC, and the mortality rate was 1.7 to 2.4 per 100,000 women [81]. According to the American Cancer Society (ACS), in 2021, there will be an estimated 66,570 new cases of the uterine body cancer diagnosed in the United States and more than 12,940 deaths [82]. These calculations include both EC and uterine sarcomas. Up to 10% of uterine body cancers are sarcomas, so the actual numbers for EC cases and deaths are slightly lower than these estimates [82].

Nowadays, worldwide, EC is the seventh most common malignant disorder, but incidence varies among regions [3]. In less developed countries, risk factors are less common and EC is rare, although specific mortality is higher. Uterine corpus cancer is the 6th leading cause of cancer death among women in the United States and the 8th leading cause of cancer-related death amongst European women [83]. The incidence is ten times higher in North America and Europe than in less developed countries; in these regions, this cancer is the most common of the female genital organs and the 4th most common site after breast, lung, and colorectal cancers [3,83].

During the past two decades, the incidence and mortality rate for EC has increased by more than 100% [80,84,85]. Moreover, the incidence varies ~10-fold worldwide, with estimated age-standardized rates of 15 per 100,000 women and higher in 2018 in Europe and North America (developed countries) [84,85]. 

EC affects mainly post-menopausal women [86]. The average age of women diagnosed with EC is 60. It is uncommon in women under the age of 45 [82].

ECs are classified into various histological subtypes, including endometrioid EC, serous EC, clear-cell EC, mixed EC, and uterine carcinosarcoma (UCS), which differ in their frequency, clinical presentation, prognosis, and associated epidemiological risk factors [82,83]. 

Most EC are adenocarcinomas, and endometrioid cancer is the most common type of adenocarcinoma [82]. Endometrioid cancers arise from the glandular cells of the endometrium, and they look like the normal endometrium. There are many variants (or sub-types) of endometrioid cancers including adenocarcinoma (with squamous differentiation), adenoacanthoma, adenosquamous (or mixed cell), secretory carcinoma, ciliated carcinoma, and villoglandular adenocarcinoma [82].

Endometrioid ECs constitute more than 80% of newly diagnosed EC cases [83]. These cancers with its subtypes are generally estrogen-dependent and have a mean age at diagnosis of 62 years [83]. In contrast, serous ECs and clear-cell ECs are relatively uncommon, accounting for ~10% and 3% of newly diagnosed ECs, are generally estrogen-independent, and are diagnosed later in life (mean of 66.5 and 65.6 years, respectively) [82,83]. 

The prognosis for most newly diagnosed EC patients is good, with a relative 5-year survival rate of 81.1% (2008–2014) [83,87]. The generally high survival rate for EC is largely driven by the frequent early detection of endometrioid ECs, coupled with the effectiveness of surgery for treating many early-stage, low-grade EECs.

### 3.2. Risk and Protective Factors of Endometrial Cancer

Multiple genetic (non-modifiable) and non-genetic (modifiable) risk factors have been associated with the development of EC (Figure 1) [78,88,89]. Genome-wide association studies have found nine independent SNPs being significantly associated with EC [52].

Race is a non-modifiable, genetic factor that plays an important role in the development of EC, as rates are highest in North America and northern Europe, lower in eastern Europe and Latin America, and the lowest in Asia and Africa [79,84,90]. Age is another non-modifiable risk factor. It is well-documented that EC primarily affects postmenopausal women, with the average age of 60 at the time of diagnosis [90]. The peak age-specific incidence is from 75 to 79 years, with 85% of cases occurring after the age of 50 and only 5% before the age of 40 [90]. Young, premenopausal women diagnosed with EC usually have other factors, contributing to the risk of the disease. 

Several non-genetic risk factors are linked with an increased risk of EC, particularly for the most prevalent histological subtype of endometrioid EC [78]. These include obesity, physical inactivity, excess of endogenous estrogens, insulin resistance, and polycystic ovary syndrome [3,78,79,84,88]. In addition, conditions involving excess of exogenous estrogens due to hormone replacement with unopposed estrogen (i.e., estrogen therapy without progesterone) predispose women to endometrial cancer [88,90]. 

Tamoxifen (selective estrogen receptor modulator (SERM)) used for breast cancer treatment approximately doubles the risk of both endometrioid and non-endometrioid types of EC if administered for 5 years and longer [78,88]. The mechanism behind is antiestrogenic effects in the breast and proestrogenic effects in the uterus [88,91]. 

The recent systematic review studying risk factors of EC concluded the presence of strong evidence associating increased body mass index (BMI) and obesity with the risk of EC development [78,85,90,92]. According to the US statistics, 57% of all ECs are attributable to obesity [76,80,86]. In the United Kingdom (UK) almost half of all ECs are attributed to overweight (BMI ≥ 25 kg/m^2^) and obesity (BMI ≥ 30 kg/m^2^) [93]. If compared with all other cancers, EC has the strongest association with obesity [78,88,93]. Women with a normal BMI have a much lower lifetime risk of EC (up to 3%), but for every 5-unit increase in BMI, the risk of EC increases by more than 50% [88,93,94]. Although the average age at diagnosis is 63 years, EC incidence is increasing among young obese women [88]. Specific lipid metabolites, including phospholipids and sphingolipids (sphingomyelins), demonstrated good accuracy for the detection of EC [93]. The underling mechanisms of the association of obesity with EC are not fully understood; however, they likely include higher estrogen levels in postmenopausal women due to aromatase activity and adipose tissue conversion of androgens into estrogens, hyperinsulinemia, and chronic inflammation [78,95,96].

As a condition closely associated with insulin resistance and obesity, highly suggestive evidence that diabetes mellitus increases the risk of EC was reported in recent systematic reviews [78,97]. Hyperinsulinemia, which is a common phenomenon prior to diabetes onset, likely has a causal association with EC through direct mitogenic effects or by increasing the levels of bioavailable estrogen through a reduction in sex hormone binding globulin levels [78,98].

However, there are some factors that have protective effect against EC [3,78,88]. Those factors include parity (with an inverse association between parity and the risk of endometrial cancer) and oral contraceptive pills [88]. The recent systematic review studying risk factors of EC found strong evidence for a 40% reduction in endometrial cancer incidence among parous compared to nulliparous women [78]. Hormonal changes during pregnancy may explain this association, usually featured by increased progesterone production with protective effects on the endometrium [78]. 

Oral contraceptive use reduces the risk of endometrial cancer up to 40%. Moreover, the longer the administration, the stronger the protective effect, which can persist even decades after cessation [88,99]. Additionally, coffee consumption has been shown to be inversely associated with EC [78,100,101].

Some researchers reported evidence that smoking reduced the risk of EC in cohort studies, although the evidence became strong when case–control studies were included [78,101]. The majority of the published cohort studies showed a reduction in risk of endometrial cancer among current or former smokers compared to never smokers [78,92,102,103]. A mechanism behind the link between decreased incidence of EC and smoking is the possible anti-estrogenic effect of nicotine; however, it has limited direct evidence and requires further investigations [78].

### 3.3. Pathophysiology of Endometrial Cancer

Based on epidemiology, histopathology, prognosis, and treatment, EC can appear as type 1 (endometrioid), affecting approximately 80% of patients, and type 2 (non-endometrioid), affecting approximately 20% of patients [78,85,88]. Type 1 tumors develop from atypical glandular hyperplasia. This type is related to long-lasting unopposed estrogen stimulation and often preceded by endometrial hyperplasia [3,90]. The molecular basis of this process is not clear yet [3]. 

Carcinomas of type 1 are associated with significant incidences of *CTNNB1*, *KRAS*, and *POLE* oncogene mutations; phosphatase and tensin homolog (*PTEN*) tumor suppressor gene; defects in deoxyribonucleic acid (DNA) mismatch repair; and near-diploid karyotype (Table 2) [3,88,104]. From a molecular point of view, ECs resemble proliferative rather than secretory endometrium [3,78]. Specific tumor suppressor gene, *PTEN* that is expressed most highly in an estrogen-rich environment, could be responsible for the disease development. Progestogens affect *PTEN* expression and promote involution of *PTEN*-mutated endometrial cells in various histopathological settings [3,78]. This hypothesis can explain therapeutic effect of progestogens in EC cases.

Type 2 tumors include predominantly unspecified EC, clear-cell, carcinosarcoma and high-grade EC, and mixed (typically endometrioid and a high-grade non-endometrioid pattern) variants [103]. Type 2 tumors are associated with mutations in TP53 and *ERBB-2 (HER2/neu)* overexpression (Table 2) [3]. The features of endometrial serous carcinomas are the following: presence of *TP53* mutations, an overall low mutation rate, and frequent copy-number alterations [88]. 

For the majority of EC cases, sporadic mutations are responsible; however, approximately 5% of EC cases are caused by inherited genetic mutations. EC caused by genetic predispositions typically occur 10 to 20 years before sporadic EC [90]. The following syndromes are known to predispose to EC: Lynch syndrome (LS), an autosomal dominant syndrome, results from a germline mutation in one of four DNA mismatch repair genes—*MLH1*, *MSH2*, *MSH6*, or *PMS2* [90]. It is associated with significantly increased lifetime risk of colorectal, ECs and some other cancers [90,105].Cowden syndrome: Cowden syndrome is an autosomal dominant syndrome characterized by *PTEN* mutations. It is associated with a 19% to 28% risk of EC by age 70 [90].

Currently, there is no approved effective screening program for EC. However, for patients with genetic syndromes, because of the significantly increased risk of the disease onset in reproductive age, the ACS recommends annual EC screening with endometrial biopsies starting at age 35 [85,90,105].

### 3.4. Clinical Presentation and Diagnostic Tools

Nowadays, for the general population there are no approved screening programs for the early detection of EC [90].

Patients’ evaluation should include thorough history taking, especially focusing on family history and possible risk factors [90]. Symptoms of EC are non-specific; thus, diagnosis of the condition is challenging in some cases. Abnormal uterine bleeding (AUB) is the most common symptom of endometrial cancer and is present in 90% of affected patients [3,84,85,90,106,107]. However, this symptom appears to be present in many other female genital disorders. Furthermore, as AUB can also be a sign of EC in premenopausal women, who comprise 20% of cases of EC, the approach to a patient with abnormal uterine bleeding will depend on the age group this patient belongs to (reproductive or postmenopausal) [79,84,85]. All postmenopausal women with AUB, especially if any of the risk factors discussed above are present [108,109], should undergo endometrial biopsy [3,79,84]. The risk of EC in postmenopausal women with uterine bleeding is up to 10% [3,84,90].

Women may also present with vague complaints of increased vaginal discharge or an incidental finding of a thickened endometrium on imaging [90]. Patients with advanced stages of the disease may complain of pelvic pain, abdominal distension, early satiety, changes in bowel or bladder function, pain during intercourse, and dyspnea because of pleural effusion [90]. However, it is important to keep in mind that up to 5% of patients with EC are asymptomatic [3,90].

Transvaginal ultrasound (TVUS) is a widely used approach for further investigations in patients suffering from AUB [3,85]. After the thorough investigation via sonography, the vast majority will undergo endometrial sampling [110,111]. The most useful approach to diagnose and confirm EC is endometrial sampling with histological examination [3,84,106,107]. 

The strategy with TVUS, followed by endometrial biopsy if an abnormality is detected, is the most cost-effective; therefore, TVUS is considered as the first step in any woman presenting with AUB [3,85,112,113]. 

Endometrial biopsy could be performed using different devices [84]. However, the most popular are the following methods: dilation and curettage (D&C), Pipelle sampling (Pipelle de Cornier prototype), and hysteroscopy with targeted biopsy. Histological examination reports may include presence of endometrial cells, atypical glandular cell of uncertain significance, or adenocarcinoma in situ [3,85,90].

## 4. Molecular Basis of Endometriosis and Endometrial Pathology

### 4.1. Genetic and Epigenetic Changes in Endometriosis

#### 4.1.1. Genetic Association and Meta-Analyses Studies

Endometriosis is a complex disease with multiple genetic and environmental factors contributing to disease pathology [114,115]. First evidence for the presence of a heritable component contributing to endometriosis came from studies published as early as the 1950s [116] that demonstrated familial clustering of endometriosis [117,118,119]. These studies showed that first-degree relatives of affected women have a five to seven times higher risk of being diagnosed with endometriosis [117,118]. Familial endometriosis was further shown to be associated with earlier age of symptom onset and a more severe disease course [120]. The genetic predisposition to endometriosis was corroborated by twin studies that showed an increased disease risk in monozygotic versus dizygotic twins, and the estimated contribution of genetic factors to endometriosis was up to 51% [121].

Large-scale genetic linkage and meta-analyses represented an important means to identify endometriosis susceptibility loci [122]. Most notable, family-based linkage studies of endometriosis conducted by the International Endogene Consortium in two combined cohorts of Australian and UK families identified two linkage regions that likely harbor rare causal variants, one on chromosome 10q26 [123] and one on chromosome 7p13–15 [124]. A third region of suggestive linkage identified by Treloar et al. is located on chromosome 20p13 [123]. Chromosome 10q26 contains two genes that were previously implicated in candidate gene mapping studies as potential endometriosis risk loci, *EMX2* [125], which encodes a transcription factor required for reproductive-tract development [126], and the tumor suppressor gene *PTEN*, which encodes a phosphatidylinositol-3,4,5-triphosphate 3-phosphatase [88]. 

Both *EMX2* and *PTEN* were previously reported to be aberrantly expressed in endometrial lesions [125,127,128,129,130]. However, systematic resequencing of the region could not confirm either gene as an endometriosis risk locus [131]. Instead, *CYP2C19* (Cytochrome P450 Family 2 Subfamily C Member 19), a nearby gene, was found to be weakly associated with endometriosis [132,133]. *CYP2C19* is a member of the cytochrome p450 family and encodes an aromatase associated with the metabolism of drugs and estrogen [134,135]. The linkage peak on chromosome 7p13–15 may represent a susceptibility allele with high penetrance for more severe forms of endometriosis [123], but the involved allele remains elusive. 

Other genome-wide association studies conducted in women of European ancestry led to the identification of two new genomic regions associated with a significant risk of endometriosis. The first locus with significant disease association was located to chromosome 7p15.2 [135]; this region may regulate expression levels of nearby gene(s) involved in the development of the uterus and endometrium [136]. A second genetic variant was mapped to chromosome 1p36.12 near the *WNT4* gene [136], which is implicated in the development and function of the female reproductive tract and sex hormone metabolism. Both risk loci were independently confirmed in Japanese and European cohorts [137,138].

Genome-wide studies identified additional susceptibility loci for endometriosis [139,140,141,142]. Several candidate genes were mapped that exhibit varying degrees of disease association including genes involved in hormone signaling (*GREB1*), cell proliferation and differentiation (*ID4*, *CDKN2PAS*), as well as cell migration and invasion (*FN1*, *VEZT*) [137,138,143]. However, most polymorphisms identified by genome-wide association studies to date are located in non-coding regions, suggesting they affect the expression of nearby genes [137,138].

In conclusion, genome-wide association studies, with few exceptions, failed to confirm a clear association between endometriosis and specific risk loci. This may indicate that there are many genetic variants, each of which has a weak impact on endometriosis development, yet in combination they can significantly increase the likelihood of endometriosis and, thus, represent true endometriosis risk loci [144,145,146]. Detection of weak effects of gene variants influencing a complex trait such as endometriosis, therefore, requires datasets of significant size.

#### 4.1.2. Genome Mapping Studies and Targeted Gene Sequencing

In addition to genome-wide association studies, candidate gene approaches were used to test the association of specific genes with endometriosis. These studies focused on systematic sequencing of identified risk loci or used genetic mapping where variants of a gene of interest with an inferred pathophysiological relevance are tested for association with the disease in samples of endometriosis cases and controls. These approaches identified genes involved in sex hormone metabolism and signaling, growth factor signaling, cell adhesion, apoptosis, cell-cycle regulation, detoxification, and inflammation [138]. Several studies also reported genetic aberrations in tumor suppression genes, such as TP53 and PTEN, in endometriotic tissues [88]. However, most associations identified by targeted gene mapping approaches suffered from low statistical power and lack of replication [145,146]. 

#### 4.1.3. Genome-Wide Sequencing Studies

Endometriosis is characterized by the growth of ectopic endometrial-like epithelium and stroma [40,41] with neoplastic characteristics that shares striking similarities with malignancy [147]. Indeed, endometriosis shares many of the key hallmarks of cancer including resistance to apoptosis, stimulation of angiogenesis, invasion, and inflammation [148]. Moreover, endometriosis is well-established as the precursor of clear cell and endometrioid ovarian carcinomas [149]. A plausible link between benign endometriosis and endometriosis associated cancer was provided by several recent next-generation sequencing approaches [121,122,123,124]. These studies also offered important insight into the molecular basis of cancer development.

Anglesio et al. were the first to report on the genome-wide identification of somatic cancer driver mutations in deep infiltrating endometriosis [150]. Deep endometriosis represents a subtype of endometriosis that occurs under the peritoneum [40] and rarely undergoes a malignant transformation. The cited authors identified somatic mutations in *PIK3CA*, *KRAS*, and *PPP2R1A*, which encodes a regulatory subunit of protein phosphatase 2. In addition, frequent loss of function mutations in AT-rich interactive domain 1A (*ARID1A*) were detected, altogether affecting approximately one-quarter of patients subjected to comprehensive genomic analysis [150]. Targeted sequencing of a subpopulation of patients further identified *KRAS* activating mutations in one-quarter of deep endometriosis patient samples [150]. Overall, of the 24 women taking part in the study, 19 had one or more driver mutations in their endometriosis tissue that were not present in their normal tissue [151,152]. Notably, cancer-associated mutations were found only in laser microdissected epithelial cells of ovarian and extraovarian pelvic endometriotic tissues, but not in stromal cells of the same tissue. These findings suggest that the occurrence of driver mutations in the epithelium is clonal and contributes to endometriosis development independently of stroma [152].

Besides *ARID1A*, *PIK3CA*, *KRAS*, and *PPP2R1A*, several other cancer-associated genes, such as *PTEN*, *PIK3R1*, *TP53*, *FBXW7*, and *CTNNB1*, were recurrently mutated in both endometriotic and uterine endometrial epithelium samples. In particular, *KRAS* and *ARID1* are frequently mutated in the endometriotic epithelium, although these epithelia were histologically benign and normal [153,154]. All of these mutations are well characterized cancer driver mutations that are known for controlling cell proliferation and survival, angiogenesis, invasion, and DNA damage repair. Importantly, besides deep endometriosis [150], other types of endometriosis also contained somatic cancer driver mutations, including endometriotic cysts, iatrogenic endometriosis as a rare complication associated with laparoscopic supracervical hysterectomy (LASH), and eutopic normal endometrial epithelium [151,155]. 

How precisely cancer driver mutations affect endometriosis in histologically normal tissue is still an outstanding question. The presence of these mutations in benign endometriotic lesions is clearly non-random. However, affected epithelial cells only carried one to two somatic mutations, which is not sufficient for malignant transformation [156]. Given the known roles of driver mutations in cancer progression, one can speculate that these mutations are necessary for driving the growth of endometriotic tissue in other regions of the body. Only accumulation of additional driver mutations in combination with microenvironmental factors, such as chronic estrogen exposure and/or inflammation, may then lead to cancer development.

### 4.2. Endometrial Stem Cells in Pathogenesis of Endometrial Pathology

There are several theories to account for the origin of endometriosis and to explain how tissue can be scattered throughout the abdominal cavity. However, there is no single theory that can explain all clinical presentations and pathological features observed in endometriosis, and several mechanisms may in fact contribute. 

The stem cell origin theory of endometriosis has gained considerable attention in recent years following the advances in molecular and genetic findings. There are two main models that are differentiated based on the tissue origin of the stem cells: stem cells arising from the regenerating uterine endometrium or stem cells originating from the bone marrow. The uterus in women is the only organ that undergoes repeated cycles of physiological damage, repair, and regeneration following menstrual shedding [114,115,116]. Menstrual shedding, and the subsequent repair of the endometrial functionalis, is a process unique to humans and higher-order primates [117,118,119]. These approximately 400 cycles of shedding and regeneration occur over a woman’s lifetime. This significant regenerative capacity is thought to be driven by stem cells that reside in the terminal ends of the basalis glands at the endometrial/myometrial interface, also termed endometrial functionalis layer, which persists after menstruation and regenerates the epithelium during the proliferative phase in response to estrogen [9]. The first model proposes that circulating epithelial progenitor or stem cells intended to regenerate the uterine endometrium are shed with menstruation and may become aberrantly activated and trapped outside the uterus, thus giving rise to ectopic lesions after retrograde menstruation and trans-tubal migration in to the pelvic cavity [157]. 

Irrespective of the site of stem cell origin, the growth of the ectopic tissue, which retains hormone responsiveness, is further influenced by sex hormones and other factors present in the microenvironment. These factors collectively control the adhesion, proliferation, angiogenesis, and invasion of the trapped progenitor cells. The ectopic tissue, in turn, induces the recruitment of immune cells leading to local inflammation and the formation of a dysregulated inflammation–hormonal autoregulatory loop. The trapped progenitor cells thereby may form nascent glands in the epithelium through clonal expansion leading to the establishment of deep infiltrating endometriosis. 

However, more studies for a better understanding of endometrial epithelial stem cell function and regulation are required to understand the eventual changes behind the endometrial pathologies.

### 4.3. Endometriosis as a Risk Factor for Endometrial Cancer

With respect to endometriosis itself as a risk factor for other conditions, women with endometriosis have a higher risk of infection, allergy, autoimmune disease, psychiatric conditions, preterm birth, metabolic syndrome, coronary heart disease, and cancer, especially ovarian [158] and breast cancers, and melanoma [159].

A history of endometriosis has been recognized as a precursor lesion of several types of malignancies and endometriosis-associated carcinoma [160,161].

Some investigations suggested no association between endometriosis and EC [28,149,160,162,163,164]. One of the recent systematic reviews performed to search for evidence on the association of endometriosis with gynecological cancers also reported no clear association between endometriosis and EC [165]. On the other hand, some studies have reported an association between endometriosis and EC reflecting overlapping risk factors between the two conditions, including endogenous or exogenous hyperestrogenism and ovulatory dysfunction [160,166]. Another recent epidemiologic study on the association of endometriosis with malignancy reported that patients with endometriosis were significantly more likely to be diagnosed with EC at a younger age than those without endometriosis (mean age at EC diagnosis 57 years vs. 62 years; *p* = 5.0 × 10^−11^) [167]. Moreover, two population-based studies have shown associations between endometriosis and EC [52,168,169]. 

If we analyze the role of risk factors in the development of endometriosis and EC, some overlapping genetic factors (Figure 1) are worth highlighting. Genetic correlation analyses by Painter et al. (2018) revealed the presence of “weak to moderate, but significant” genetic overlap between endometriosis and EC [52,144]. Namely, in the cross-disease meta-analysis the authors found 13 SNPs that appeared to be involved in replication. These SNPs are the following: rs2475335, rs9865110, rs2278868, rs12303900, rs9349553, rs10008492, rs9530566, rs10459129, rs2198894, rs7042500, rs17693745, rs7515106, and rs1755833 [52,144]. SNP rs2475335, which is located on chromosome 9p23, was most significantly associated with both diseases (*p* = 4.9 × 10^−8^) [52].

To conclude about the link between endometriosis and EC, epidemiological studies have reported conflicting data for an association between the diagnosis of endometriosis and risk of EC [52]. More large-scale investigations are required in order to confirm or refute the link between endometriosis and risk of EC development. 

## 5. Clinical Applications of Current Knowledge and Directions for Future Research

### 5.1. Molecular Basis for a Specific Therapeutic Approach

Collectively, studies performed over the last decade shed new light on the pathophysiology of endometriosis. Linkage and sequencing studies have identified genes and pathways important for endometriosis development and have highlighted potential causal links between endometriosis and endometriosis-associated cancer. The identification of women with endometriosis who are at risk of cancer development provides a basis for improved diagnosis and prognosis and is likely to aid in improved cancer surveillance of patients at risk. 

As treatment of endometrial cancer was based on histological characteristics and staging [170,171], prognosis was not promising, especially if the stage is advanced [172,173,174]. Therefore, in the last couple of years, efforts have been directed to molecular aberrations within the specific tumor, as a novel biological targeted therapy with promising outcomes in clinical trials [175]. 

Various biomarkers [176], such as *mTOR* pathway disruptions, loss of estrogen and progesterone nuclear expression, TP53 mutation, changes in Wnt-signaling, or *L1CAM* expression, were identified as a link to endometrial cancer development [177]. All these mutations were associated with poor prognosis, but their clinical utilization is still questionable. 

Preclinical investigations related to molecular-targeted therapies in ECs enabled deeper understanding of underlying mechanisms and highlighted different approaches to EC patients. 

According to genomic characteristics of 373 endometrial carcinomas, The Cancer Genome Atlas (TCGA) classified EC into four molecular subtypes [178,179], which differ a lot from the molecular point, underlying risk factors, clinical and pathological features, treatment modalities, and prognosis [180,181,182]. These four distinct prognostic groups are *POLE* ultramutated, microsatellite instability/hypermutated, copy number-low microsatellite stable, and copy number-high/serous like.

Compared with other subtypes of tumors, prognosis for the copy number-high/serous-like group of patients is poor [183]. Poor prognosis is related to the loss of tumor suppressor *TP53* resulting in a high degree of genomic instability and rapid tumor progression and invasion [184,185]. As one-quarter of serous-like tumors have ERBB2 overexpression, there is a need to investigate the role of human epidermal growth factor receptor 2 (HER2)-targeted inhibitors [186,187,188]. Considering molecular similarities between high-grade endometrioid and serous carcinomas, patients of this subtype may benefit from treatment as if their tumor was serous. Moreover, specific mutations and overexpression of molecular targets in these tumors could tailor treatment in both the primary and recurrent setting.

From the clinical point of view, there is a need to create an integrated molecular risk profile for endometrial cancer. For that purpose, these four molecular subgroups were combined with additional molecular markers. Integrated molecular and clinico-pathologic risk assessment was based on a multivariate analysis of four molecular subgroups, clinical and histopathological characteristics of tumors, and various molecular classifiers. Molecular markers involved were *TP53* expression, *MSI*, *POLE* mutation, protein expression of *L1CAM*, *ARID1a*, *PTEN*, *ER/PR*, as well as analysis of 13 genes found to have variable expression in the TCGA classification groups (*BRAF*, *CDKNA2*, *CTNNB1*, *FBXW7*, *FGFR2*, *FGFR3*, *FOXL2*, *HRAS*, *KRAS*, *NRAS*, *PIK3CA*, *PPP2R1A*, and *PTEN*). This integrated model for prediction of endometrial cancer recurrence was confirmed to be more reliable than the traditional one relying on clinical and pathologic factors [189]. Moreover, this classification system enhances risk stratification of endometrial cancers. Importance for molecular subtyping was confirmed in clinical practice, as sorting of patients into molecular subgroups was confirmed to predict response rates to conventional, targeted systemic and radiotherapy [190,191]. For clinicians, molecular subtype stratification could be used in both preoperative evaluation (whether to prepare the whole set up for lymph node dissection or not) and postoperative treatment (the need for eventual adjuvant therapy). Moreover, molecular subtyping was confirmed to be very important for targeted therapy in patients with recurrent and metastatic diseases [175,177,192,193,194]. Thus, once the diagnosis of endometrial cancer has been established, there is a need to perform molecular subtyping, which will enable proper therapeutic approach.

### 5.2. Prognostic Biomarkers for Endometrial Cancer

There is no screening method for EC for the general population. Women with LS and their first-degree relatives are offered annual screening with TVUS and endometrial biopsy from the age of 35 years [90,105].

There are several types of biomarkers: gene-based biomarkers, proteins biomarkers, and hormonal biomarkers [105]. Gene-based biomarkers include the following: *PTEN*, *TP53*, microribonucleic acids (microRNAs), circulating tumor DNA, and DNA methylation. Protein biomarkers include *pRb2/p130*, *Ki 67*, *ARID1A*, cell adhesion molecules (CAMs), phosphohistone-H3 (*pHH3*), angiotensin factors, etc. The most commonly mutated genes detected in EC patients using Tao brush samples were *PTEN* and *TP53* [83]. 

One of the gene-based biomarkers, *PTEN* tumor suppressor, antagonizes the phosphoinositol-3-kinase/AKT signaling pathway, suppressing cell survival as well as cell proliferation [105]. A recent study suggests that *PTEN* expression in endometrial hyperplasia can be used as an early warning of heightened cancer risk [105,195]. Complete loss of *PTEN* protein expression is most commonly found in EC and endometrial hyperplasia with cytological atypia. 

Another potentially useful molecular biomarker is *TP53*, which belongs to cell cycle proteins [105], and triggers cellular responses that can lead to cell-cycle arrest, senescence, differentiation, DNA repair, apoptosis, and inhibition of angiogenesis [196]. The role of *TP53* in EC and hyperplasia has been studied, showing that TP53 gene mutation is present in EC, but it is absent in endometrial hyperplasia [105,197]. 

The expression of the cell cycle regulator *pRb2/p130* was evaluated in EC and endometrial hyperplasia and was found to be highly expressed in the proliferative endometrium and in hyperplasia without atypia, but it was downregulated in secretory endometrium, atypical hyperplasia, and EC [105,197]. 

The most promising serum biomarker for EC is human epididymis protein 4 (HE4) [105]. A number of studies have looked at HE4 as a prognostic marker for EC [86,105,198,199]. Diagnostic levels range between 50 and 70 pmol/L, with a minimum 78% sensitivity and 100% specificity, even in early-stage disease [105,198]. Serum HE4 levels are found to be significantly higher in advanced stages of EC [105,199] and are predictive for disease recurrence [105]. 

There are many novel biomarkers under investigation. Introduction of them into clinical practice could improve timely EC diagnosis, treatment outcomes, and surveillance of EC patients. 

## 6. Conclusions

Although diagnostic methods of endometriosis are well-developed in modern gynecology, the etiopathogenesis of the disease remains largely unknown. Lack of clear understanding of the pathologic process leads to inferable outcomes in patients suffering from endometriosis and may be linked to development of related female genital malignancy. Existing studies have reported inconclusive data for an association between endometriosis and risk of EC. Further large-scale investigations could help to answer this query. Molecular studies of endometrial tissue function and endometriosis might shed light on the real cause of the condition and the factors leading to EC development. As a result of a better understanding of the molecular basis of endometriosis and EC, patient management and outcomes could be improved. 

## Figures and Tables

**Figure 1 ijms-22-09274-f001:**
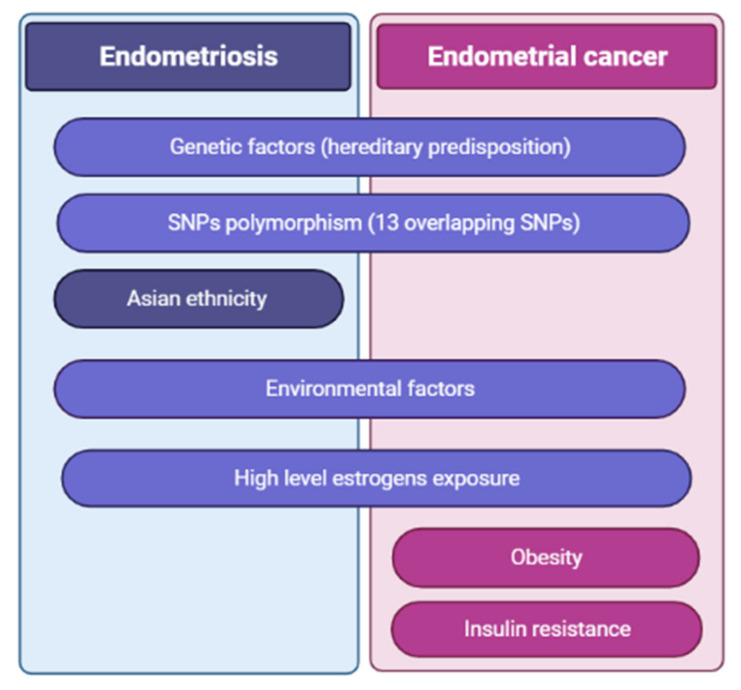
Risk factors for endometriosis and endometrial cancer. Created at BioRender.com (accessed on 15 July 2021).

**Table 1 ijms-22-09274-t001:** Classification of endometriosis.

Classification						
***American Society for Reproductive Medicine (ASRM)***						
***Staging***				***Points***		
Stage 1—Minimal Endometriosis				1–5		
Stage 2—Mild Endometriosis				6–15		
Stage 3—Moderate Endometriosis				16–40		
Stage 4—Severe Endometriosis				>40		
***ENZIAN (supplement to ASRM)***						
*Compartments*						
Compartment A: vagina, recto-vaginal septum;						
Compartment B: uterosacral ligaments to the pelvic wall (BB: bilateral involvement);						
Compartment C: rectum and sigmoid colon.						
					*Disease severity*	*Invasion*
					Grade 1:	<1 cm
					Grade 2:	1–3 cm
					Grade 3:	>3 cm
						*Deep endometriosis invasion beyond the pelvis*
						FA: adenomyosis
						FB: bladder invasion
						FU: intrinsic ureteral endometriosis;
						FI: bowel disease cranial to the sigmoid colon
						F0: other locations
***The Endometriosis Fertility Index (EFI)***						
*Historical factor*		*Years*	*Points*			
Patient age		≤35	2			
		36–39	1			
		≥40	0			
Duration of infertility		≤3	1			
		>3	0			
Prior pregnancy		Yes	1			
		No	0			
					*Score*	*Description*
					4	Normal
					3	Mild
					2	Moderate
					1	Severe
					0	Absent or nonfunctional

**Table 2 ijms-22-09274-t002:** Molecular mechanisms of endometrial cancer development.

Endometrial Cancer Type	Molecular Factors/Genes	Changes in Function Leading to Endometrial Cancer
Type 1		
	*CTNNB1*	Mutation
	*POLE*	Mutation
	*KRAS*	Mutation
	*PTEN*	Loss
	*AKT*	Up-regulation
	*PI3KSA*	Up-regulation
	G_1_/S cell cycle phase	Progression
	*Bcl-2*	Loss of down-regulation
	*MLH1/MSH6*	Instability
	DNA	Mismatch
Type 2		
	*TP53*	Mutation
	*ERBB-2 (HER2/neu)*	Overexpression
	*p-16*	Inactivation
	*E-cadherin*	Reduction

## Data Availability

No raw data to share.

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
