# Peer review of "Molecular Basis of Endometriosis and Endometrial Cancer: Current Knowledge and Future Perspectives"

_ijms, 2021, doi:10.3390/ijms22179274_

Round 1

Reviewer 1 Report

Good evening! 

I have read your work and I must say that it is an easy to read and follow article with very good English use.However, unfortunately I cannot find valuable or novelty pieces of information or research in the article, as the information about endometriosis and endometrial cancer that you have displayed are very general and unspecific and already well known and studied and are mostly just general facts that clinicians are well aware of and I do not find this work to bring significant contribution to the field.

Author Response

Good evening!

I have read your work and I must say that it is an easy to read and follow article with very good English use.However, unfortunately I cannot find valuable or novelty pieces of information or research in the article, as the information about endometriosis and endometrial cancer that you have displayed are very general and unspecific and already well known and studied and are mostly just general facts that clinicians are well aware of and I do not find this work to bring significant contribution to the field.

Response:

Dear Reviewer,

Thank you very much for the review of our manuscript. We appreciate your time, efforts, and valuable comments. The manuscript is prepared as a non-systematic review summarizing existing data on endometriosis and endometrial cancer, and looking for a link between these conditions on a molecular level.  We tried to offer a complete and comprehensive overview of the topic for the readers, in order to allow a clear counselling also for the clinicians during the management of women affected by endometriosis.

Reviewer 2 Report

This is an informative manuscript summarizing the pathology of endometriosis and endometrial cancer. It also provides an overview of existing research on the development of these two conditions on a molecular level. Additionally, it offers useful clinical recommendations. My suggestions for the articles are as follows.

  1. Conclusions should be Section 6, not 7.
  2. The manuscript would benefit from a lot more figures illustrating the information in the article. For example, there could be a picture on the histology of endometrium and endometriosis, a table for the different classifications of endometriosis, etc.
  3. In line 51-52, the author wrote “The 3D imaging revealed a more complex network of endometrial glands in human endometrium.” Please be more specific with the description. More complex compared to what? 2D structure? Similarly, please be more specific as to what condition the following sentence is describing: The plexus structure of the glands during menstruation remained at the bottom of the endometrium and crept along the muscular layer and it was mainly located in the stratum basalis regardless of the menstrual cycle phase.
  4. Please explain the differences between adenomyosis and endometriosis early on before introducing the classification of adenomyosis.
  5. Please add details to Figure 1 legends, such as what the color and size of each bar means, what the 13 overlapping SNPs are, etc. For Figure 2, I recommend making a table listing reported molecular mechanisms and a summary for each mechanism on how it has been shown to be involved in endometrial cancer development.
  6. The article needs to be extensively edited by a life sciences editor who is a native English speaker to fix issues with grammar and sentence structures.

Author Response

Dear Reviewer,

Thank you very much for the detailed review of our manuscript. We appreciate your time, efforts, valuable comments and suggestions that helped us to improve the text quality. Please find below our point-by-point responses for all your comments.

  1. Conclusions should be Section 6, not 7.

Response: Thank you for the comment. This typo mistake was corrected. Conclusion is numbered as section 6 in the text.

  1. The manuscript would benefit from a lot more figures illustrating the information in the article. For example, there could be a picture on the histology of endometrium and endometriosis, a table for the different classifications of endometriosis, etc.

Response:  Thank you for the comment.

Following the Reviewer comment, a table summarizing classifications of endometiosis was created.

Table 1. Classifications of endometriosis

Classification

American Society for Reproductive Medicine (ASRM)

Staging

Points

Stage 1 - Minimal Endometriosis

1-5

Stage 2 - Mild Endometriosis

6-15

Stage 3 - Moderate Endometriosis

16-40

Stage 4 - Severe Endometriosis

>40

ENZIAN (supplement to ASRM)

Compartments

Compartment A: vagina, recto-vaginal septum;

Compartment B: uterosacral ligaments to the pelvic wall (BB: bilateral involvement);

Compartment C: rectum and sigmoid colon.

Disease severity

Grade 1: invasion <1 cm;

Grade 2: invasion 1-3 cm;

Grade 3: invasion >3 cm

Deep endometriosis invasion beyond the pelvis

FA: adenomyosis

FB: bladder invasion

FU: intrinsic ureteral endometriosis;

FI: bowel disease cranial to the sigmoid colon

F0: other locations

The Endometriosis Fertility Index (EFI)

Historical factor

Patient age

≤35 years old

36-39 years old

≥40

years old

Duration of infertility

≤3 years

>3 years

Prior pregnancy

Yes

No

Score

Description

4

Normal

3

Mild

2

Moderate

1

Severe

0

Absent or nonfunctional

Table legend:

F-  "far" - referring to distant retroperitoneal structures.

  1. In line 51-52, the author wrote “The 3D imaging revealed a more complex network of endometrial glands in human endometrium.” Please be more specific with the description. More complex compared to what? 2D structure? Similarly, please be more specific as to what condition the following sentence is describing: The plexus structure of the glands during menstruation remained at the bottom of the endometrium and crept along the muscular layer and it was mainly located in the stratum basalis regardless of the menstrual cycle phase.

Response:  Dear Reviewer, thank you for this comment. The sentences starting from lines 51-52 have been rewritten and are present as follows:

“… The 3D imaging revealed a more complex network of endometrial glands in human endometrium than it was observed with a traditional 2 dimensional (2D) imaging [4]. Using 3D imaging technique, Yamaguchi and co-authors (2021) have found specific morphological features of the human endometrial glands, including occluded glands, the plexus of the basal glands, and the gland-sharing plexus with other glands, which were not sufficiently observed in the past by 2D histological methods [4]. The 3D analysis through the endometrial layers performed by the researchers clarified that the plexus structure of the glands expanded horizontally along the muscular layer. Furthermore, these morphological features were detected regardless of age or phase of the menstrual cycle, suggesting that they are basic components of the normal human endometrium [4]. These novel findings suggest that 2D layers histology, which is in use for more than 100 years, does not reflect real morphology of the endometrium.  More clear picture of the human endometrium structure could help in understanding of a different endometrial conditions etiology such as endometriosis and endometrial cancer. These diseases affect reproductive age women’s goals and quality of lives [16-18]. Understanding of the pathogenesis, immunohistochemical and molecular mechanisms of these conditions could improve patients’ management [19-22].”

  1. Please explain the differences between adenomyosis and endometriosis early on before introducing the classification of adenomyosis.

Response: Dear Reviewer, thank you for this comment. The definition of Adenomyosis was included in the text:

Adenomyosis, as “internal” uterine endometriosis, characterized by the presence of endometrial glands and stromas within the myometrium that causes myometrial inflammation and hypertrophy [34,36,37].”

  1. Please add details to Figure 1 legends, such as what the color and size of each bar means, what the 13 overlapping SNPs are, etc. For Figure 2, I recommend making a table listing reported molecular mechanisms and a summary for each mechanism on how it has been shown to be involved in endometrial cancer development.

Response: Dear Reviewer, thank you for this comment. Figure 1 has been corrected and made more clear. The figure legend and the information on particular 13 SNPs are included.

Figure legend:

Bars in blue: shared risk factors for endometriosis and endometrial cancer;

Bar in dark blue: risk factor of endometriosis;

Bars in pink: risk factors of endometrial cancer;

Information about SNPs is included in the text.

“Some genetic factors found to serve as risk factors for endometriosis. In particular, nineteen single-nucleotide polymorphisms (SNPs) were found to be associated with endometriosis [51]. The authors found a significant genetic overlap between endometriosis and EC in the genetic correlation analysis, which found 13 SNPs that appeared to be involved in development of both conditions [51].”

“Genetic correlation analyses by Painter et al. (2018) revealed the presence of “weak to moderate, but significant”, genetic overlap between endometriosis and EC [51]. Namely, in the cross-disease meta-analysis the authors found 13 SNPs that appeared to be involved in replication [168]. These SNPs are the following: rs2475335, rs9865110, rs2278868, rs12303900, rs9349553, rs10008492, rs9530566, rs10459129, rs2198894, rs7042500, rs17693745, rs7515106, and rs1755833 [51]. SNP rs2475335, which is located on chromosome 9p23, was most significantly associated with both diseases (P= 4.9 x 10-8) [51].”

Following the Reviewer suggestion, Figure 2 was removed and replaced with the table, explaining molecular mechanisms for each type of endometrial cancer.

Table 1. Molecular mechanisms of endometrial cancer development.

Endometrial cancer type

Molecular factors/genes

Changes in function leading to endometrial cancer

Type 1

CTNNB1

Mutation

POLE

Mutation

KRAS

Mutation

PTEN

Loss

AKT

Up-regulation

PI3KSA

Up-regulation

G1/S cell cycle phase

Progression

Bcl-2

Loss of down-regulation

MLH1/MSH6

Instability

DNA

Mismatch

Type 2

TP53

Mutation

ERBB-2 (HER2/neu)

Overexpression

p-16

Inactivation

E-cadherine

Reduction

  1. The article needs to be extensively edited by a life sciences editor who is a native English speaker to fix issues with grammar and sentence structures.

Response: We are grateful for the suggestion. The new version of the paper was revised by a native English speaker, as recommended.

Round 2

Reviewer 1 Report

Congrats for the updated version of the manuscript.

I find it much more appealing and scientifically sound.